# Contribution of Retrotransposons to the Pathogenesis of Type 1 Diabetes and Challenges in Analysis Methods

**DOI:** 10.3390/ijms24043104

**Published:** 2023-02-04

**Authors:** Anja Štangar, Jernej Kovač, Robert Šket, Tine Tesovnik, Ana Zajec, Barbara Čugalj Kern, Barbara Jenko Bizjan, Tadej Battelino, Klemen Dovč

**Affiliations:** 1Division of Pediatrics, University Medical Centre Ljubljana, 1000 Ljubljana, Slovenia; 2Department of Pediatrics, Faculty of Medicine, University of Ljubljana, 1000 Ljubljana, Slovenia

**Keywords:** retrotransposons, type 1 diabetes, type 1 diabetes pathogenesis, viral infections, single-cell sequencing, retrotransposons analysis methods

## Abstract

Type 1 diabetes (T1D) is one of the most common chronic diseases of the endocrine system, associated with several life-threatening comorbidities. While the etiopathogenesis of T1D remains elusive, a combination of genetic susceptibility and environmental factors, such as microbial infections, are thought to be involved in the development of the disease. The prime model for studying the genetic component of T1D predisposition encompasses polymorphisms within the *HLA* (human leukocyte antigen) region responsible for the specificity of antigen presentation to lymphocytes. Apart from polymorphisms, genomic reorganization caused by repeat elements and endogenous viral elements (EVEs) might be involved in T1D predisposition. Such elements are human endogenous retroviruses (HERVs) and non-long terminal repeat (non-LTR) retrotransposons, including long and short interspersed nuclear elements (LINEs and SINEs). In line with their parasitic origin and selfish behaviour, retrotransposon-imposed gene regulation is a major source of genetic variation and instability in the human genome, and may represent the missing link between genetic susceptibility and environmental factors long thought to contribute to T1D onset. Autoreactive immune cell subtypes with differentially expressed retrotransposons can be identified with single-cell transcriptomics, and personalized assembled genomes can be constructed, which can then serve as a reference for predicting retrotransposon integration/restriction sites. Here we review what is known to date about retrotransposons, we discuss the involvement of viruses and retrotransposons in T1D predisposition, and finally we consider challenges in retrotransposons analysis methods.

## 1. Introduction

Type 1 diabetes (T1D) is one of the most common chronic diseases of the endocrine system in childhood and adulthood, with onset usually occurring in childhood. Symptoms associated with the onset of the disease are polydipsia, polyphagia, and polyuria, along with overt hyperglycemia [1,2]. The features most useful in discrimination of type 1 diabetes include younger age at diagnosis (<35 years) with lower BMI (<25 kg/m^2^), unintentional weight loss, ketoacidosis, and glucose > 360 mg/dL (20 mmol/L) at presentation [3]. T1D is an autoimmune-mediated disease characterized by the destruction of the insulin-producing beta cells of the pancreas [1,2]. The result is a loss of blood glucose regulation, so people with T1D require lifelong daily exogenous insulin replacement therapy [1,4]. Incidence rates vary by geographic region and in 2021 ranged from a minimum of 2.51 per 100,000 in east Asia and the Pacific to a maximum of 25.06 per 100,000 in North America (data for individuals younger than 20 years) [5]. The global annual occurrence of T1D (128,900 new cases in the under 20 years age-group) was increasing by about 3% each year until 2020 but, after the pandemic COVID-19 outbreak in 2020, the annual occurrence appears to have increased by 57% [6,7,8]. The percentage of patients who presented with diabetic ketoacidosis (DKA) also increased [8]. Microbial infections (including Coxsackie B virus (CVB), rotavirus, influenza viruses, and Mycobacterium avium subspecies paratuberculosis (MAP)) are environmental factors in the pathogenesis of T1D, implying a seasonal pattern in the onset of T1D with an increase in cases in late fall, winter, and early spring [7,9,10]. Other factors thought to be involved in the development of the disease include microbiota, diet, and genetic susceptibility [9]. In addition, stress is a factor that may modulate the immune system (Figure 1). Breastfeeding and soiling are early life factors reported to have a protective effect [11]. However, when all factors are taken together, the pathogenesis of the disease is still not fully understood. The genetic component of T1D predisposition involves polymorphisms within the major histocompatibility complex region encoding human leukocyte antigen (HLA) [9]. This region (~4 Mb) is located on chromosome 6p21 and includes approximately 160 coding genes in three distinct structural regions: class I, II, and III. HLA molecules are key elements for the specificity of antigen presentation to lymphocytes [12]. The major loci associated with T1D are HLA-DR and HLA-DQ. HLA class II DR3-DQ2 and DR4-DQ8 haplotypes of HLA-DRB1, HLA-DQA1, and HLA-DQB1 genes increase the likelihood of developing the first islet autoantibody [9]. The complement C4 region is also thought to contribute to HLA-DQ-mediated disease risk [13]. In addition, some class I alleles have an independent effect on susceptibility to T1D. HLA-A*24, B*18, and B*39 accelerate progression to clinical T1D [9]. Overexpression of HLA class I in the islet during T1D development contributes to the recruitment of autoreactive CD8+ T cells that selectively target β-cells. Interferon-α (IFNα) is expressed in islet cells of T1D patients, and its expression and signalling are regulated by T1D genetic risk variants and viral infections associated with T1D [14].

Apart from polymorphisms within the HLA region, genomic reorganization, caused by repeat elements and endogenous viral elements (EVEs), might be involved in T1D predisposition. An EVE is a DNA sequence derived from a virus, and present within the germline of a non-viral organism [15]. They have been shown to be important players in genome stability and can provoke genomic rearrangements, some of which have already been associated with various diseases and pathological conditions, such as cancer, neurodegenerative diseases, autoimmunity, viral infections, and inflammation [16,17,18,19]. In T1D, they may represent the missing link between genetic susceptibility and environmental factors (Figure 1), because it has been shown that viruses can activate the expression of some EVE families [20], and this phenomenon has also been proposed for coronavirus [21]. Therefore, in this review, we aim to summarize what is known to date about EVEs, with focus on retrotransposon subclass, discuss the involvement of retrotransposons in T1D predisposition, and consider the challenges in retrotransposon analysis methods.

## 2. Reported Relationship between the Occurrence of T1D and Viral Infections

Viruses may cause damage to pancreatic β-cells via two main mechanisms. One may be direct cytolysis of virus-infected β-cells without the involvement of the immune system, and the other may be viral induction of autoimmune phenomena [4]. After viral infection, the initial response of β-cells to INFα is characterized by chromatin remodeling followed by changes in transcriptional and translational regulation [14].

The most studied viruses in relation to the onset of T1D disease and viral infections emphasize the effect of enteroviruses [20]. As a recent study of early life factors found no association with age of first school or preschool attendance, which is a marker of respiratory infections, enteroviruses may be more important [11]. One such example is CVB, whose viral RNA and antibodies against it have been detected in blood, stool, and pancreas before the development of autoantibodies (AAb) in T1D [20]. Rotavirus is another virus commonly associated with T1D pathology. As the availability of rotavirus vaccines increases worldwide, studies have investigated whether vaccination affects the development of T1D. However, no association has been found with increasing or decreasing the risk of T1D [22,23].

In a study by Labombarde et al., the induction of broadly reactive influenza antibodies (Abs), which form after repeated infection and vaccination with influenza virus subtypes, enhanced both the formation of autoreactive Abs and susceptibility to autoimmune disease. These Abs have the potential to contribute to pathogenesis associated with tolerance defects or additional assaults on the immune system. Normally, various tolerance mechanisms prevent autoreactive B cells from causing disease unless other checkpoints fail and defects in the self-tolerance lead to the formation of AAbs. Broadly reactive influenza Abs exhibit a high degree of lipid binding, a property that enhances their binding to viral epitopes. However, this property may also promote binding to self-proteins. The important observation was that they also bind various other endogenous proteins, including insulin [24].

The current worldwide epidemic of COVID-19 suggests a possible involvement of SARS-CoV-2 in the development of T1D, but there is no consensus on whether the virus can directly infect and damage pancreatic islets [25,26]. In addition, the global health campaign during the pandemic, which stressed the importance of careful hand hygiene, may have contributed to the increased incidence of T1D. Indeed, the hygiene hypothesis states that a lack of microbial exposure in early life predisposes to T1D [27].

## 3. Endogenous Viral Elements

EVEs are typically 100 to 10,000 bp long DNA sequences with the ability to change their position within a genome [28]. Nearly half of the human genome is occupied by more than three million sequences derived from EVEs. However, only a small fraction of them encodes biochemically active proteins or complex noncoding regulatory sequences that promote their transposition in both the germline and soma [28,29]. Although EVE-imposed gene regulation can be beneficial, it is usually detrimental to the host because EVEs represent a major source of genetic variation and instability. Therefore, most of them are highly regulated, usually methylated, to maintain a heterochromatic repressed state [30,31]. They are divided into two main classes based on their transposition intermediates: retrotransposons and DNA transposons, collectively known as transposable elements (TEs) [28,32]. Most DNA transposons are mobilized via a DNA intermediate through a cut-and-paste mechanism. In contrast, retrotransposons are mobilized via a copy-and-paste mechanism, in which an RNA intermediate is reverse-transcribed into a cDNA and integrated elsewhere in the genome [28]. In this review, we focus on the latter. Such elements are long terminal repeat retrotransposons, which include human endogenous retroviruses (HERVs), and non-LTR retrotransposons, which include long and short interspersed nuclear elements (LINEs and SINEs) [31]. LTR negative elements are also SVAs, which are named after the sum of their parts SINE, VNTR (variable number of tandem repeats), and Alu [19]. DNA transposons account for approximately 3% of total genomic DNA, ERVs 8%, LINEs 18%, and SINEs 14%, whereas only 2% of the human genome is occupied by protein-coding genes (Figure 2). The remaining 55% is a dark region containing genes whose activity is restricted to transcription, structures with currently unknown function, and also viral sequences [33].

The defining characteristic of EVEs is vertical inheritance. In addition, horizontal transmission of EVEs between species occurs and is an important factor in their long-term success [28]. ERVs are remnants of ancient exogenous retrovirus infections of the germline [34]. EVEs mostly accumulate in introns and intergenic regions because integration within the transcription unit is deleterious and therefore subject to negative selection and elimination during evolution [35]. Retrotransposons that can move autonomously encode the enzymatic machinery necessary for their transposition [17,36]. Functional LINE-1 elements are 6–7 kb long and usually contain two coding open reading frames (ORF1 and ORF2) and a long 5’ end with RNA polymerase II promoter activity [36,37]. ORF1 proteins are involved in the recognition and transport of the RNA template into the nucleus, and ORF2 encodes endonuclease and reverse transcriptase [36]. The promoters of active LINE-1 are most active in the germline and early embryo [38]. In reality, unrestricted ‘egoistic’ retrotransposition is rare. In the case of LINE-1 elements, most copies are inactive and often represented only by short 3’ end fragments [32,39]. They are unlikely to be important for further self-propagation, but they do generate insertional mutations [32]. Non-autonomous elements, on the other hand, are generally noncoding but contain sequence features that are recognized by autonomous retrotransposon proteins and enable their *trans*-mobilization [17,36]. For example, SINEs and SVA elements are usually *trans*-mobilized by the LINEs machinery [36,40]. The non-autonomous elements may differ significantly from the autonomous LINE-like copies. For example, LINE-1 elements are AT-rich, whereas Alu elements are GC-rich. This difference in base composition may have been selected to target different chromosomal regions [29]. The insertions are processed transcripts: intron is spliced out and the 3’ end of the inserted LINE-1 and SVA sequences are polyadenylated, whereas Alu insertions have shorter A tails [41]. Active elements and their insertion products are collectively referred to as “transposable and interspersed repetitive elements (TIREs)” [32,42]. It is worth mentioning that LINE-1 can also mobilize protein-coding RNAs [17]. 

Autonomous HERV elements are more complex and contain a minimal set of three distinct genes expressed as a single RNA: *gag* (group antigens), *pol* (polymerase), and *env* (envelope) (Figure 3) [43,44]. In addition, the *env* gene can form splice variants that generate different proteins [45]. After reverse transcription, the cDNA product is integrated into the host chromosome by a process similar to that of cut-and-paste transposases [36]. HERVs are flanked by two long terminal repeats (LTRs) that contain the regulation part of the retrovirus. However, most HERVs are present as solo LTRs [46]. The most studied HERV group is the W group, which is specifically mobilized by the LINE-1 machinery [43]. HERV-K retrotransposition also depends on the integration machinery via the 3’ poly(A) tail of the RNA, similar to LINE-1. Several transcription factors, including MITF, MZF1, NF-Y, GATA-2, and OCT3/4, are required for HERV-K LTR activation. HERV-K gene expression is synergistically induced by DNA hypomethylation and SOX2 expression. HERV-K LTR integration sites have also been detected within HLA region 6p21.32 [47].

In the UCSC Genome Browser [48], we searched the major HLA loci related to T1D (HLA region 6p21.32) for Alu, LINE-1, and ERV families of retrotransposons (Table 1). These retrotransposons were annotated by RepeatMasker, a program that searches DNA sequences for interspersed repeats [49,50]. 

## 4. The Influence of Retrotransposons on Diseases

Genomics and large-scale functional assays have revealed the extent of the influence of EVEs on genome evolution, function, and disease [51]. Most of the disease-causing insertions inactivate gene function through insertional mutagenesis or aberrant splicing. Another important mechanism of LINE-1 mediated insertions is target-site deletion [17]. In addition, retrotransposon insertions can provide the raw material for the emergence of protein-coding genes and non-coding RNAs that can take on important and, in some cases, essential cellular functions. Transposition of retrotransposons can lead to rearrangements in the genome and alter transcriptional activities by introducing *cis*- and *trans*-regulatory DNA elements [51]. Other mechanisms by which novel LINE-1, Alu, and SVA element insertions can disrupt gene function are related to epigenetic changes at the site of integration [17]. In addition, retrotransposons can promote genomic structural changes, long after they have lost the ability to mobilize [51]. Environmental stimuli, such as infection and cellular stress, as well as natural cellular processes, such as senescence, destabilize epigenetic factors that normally silence the majority of EVEs in the genome, triggering their sporadic transcriptional activation [31]. Recently, Iouranova et al. found that the KRAB zinc finger protein ZNF767 controls the transcriptional influence of LTR12-related ERVs sequences [52]. While loss of regulatory control at EVEs may be a rare stochastic event occurring in a small subset of cells, it is possible that this process is favored by selection during cell evolution. This process would favor clonal proliferation of cells in which the EVE is unmasked, and which perpetuate autoimmune response [31]. More than one hundred EVE insertions have been causally linked to Mendelian disorders and hereditary cancers [17]. Expression of retrotransposons is deregulated in many viral infections, including in COVID-19 patients. For example, LINE-1 ORF1p and ORF2p proteins have peptides identical to the SARS-CoV-2 epitope, which are targeted by Abs in COVID-19 and thus could induce an autoimmune loop by molecular mimicry with build-up of autoreactive CD4+ Th cells [18]. Another hypothesis is that coronavirus infection may increase retrotransposon expression through enhancing global demethylation activity [21].

Overexpression of ERV envelope proteins, as seen in the brains of patients with some neurodegenerative and autoimmune diseases, could trigger a variety of cellular processes and abnormalities associated with these pathologies, such as neurodegeneration, autoinflammation, demyelination, and superantigen activity [16]. Highly defective HERV *env* genes and alternative *env* splice variants may represent additional mechanisms of pathogenesis [45]. In addition, cytoplasmic accumulation of nucleic acids derived from activated retrotransposons, including double-stranded RNA, reverse transcribed cDNA, or RNA-DNA hybrids, are increasingly viewed as potent immunological ‘adjuvants’ that can trigger autoimmune responses (Figure 4) [16]. HERV proteins may act as superantigens (SAg) that trigger nonspecific polyclonal activation of autoreactive T lymphocytes and cause massive release of cytokines (including interferons) [43,53]. In addition to the direct immunogenic effects of retroviral products, HERV proteins can affect the host immune response in additional ways, for example, by transactivating/suppressing genes involved in immunomodulation. The HERV-W group has been intensively studied for its putative role in various diseases, such as cancer, inflammation, and autoimmunity [43]. Expression of proinflammatory cytokines has been shown to be induced in human monocytes after in vitro stimulation with recombinant HERV-W Env protein, a process that requires TLR4 receptor activation, whereas human dendritic cells treated with multiple sclerosis-associated retrovirus (MSRV)-Env started to promote Th1-like lymphocyte differentiation [54].

## 5. The Influence of Retrotransposons on T1D

HERVs are an important group of viral entities, which may be associated with T1D and other autoimmune diseases etiology. In T1D triggering monocytes, HERVs have been shown to induce the production of pro-inflammatory cytokines, such as IL-1, IL-6, and TNF-α [55,56]. HERVs can be transactivated by environmental viruses, such as enteroviruses, by some bacteria, and by inflammatory stimuli [4,20,57]. CVB4, the enterovirus most frequently mentioned in the context of T1D pathology, was recently found to induce the transcription of a HERV-W-*env* in primary cell cultures, such as monocytes, macrophages, and pancreatic cells [20]. It has also been shown that infection with MAP affects HERV-W Ag expression and increases AAbs production in T1D [58]. Anti-HERV-W-Env Abs were detected in the sera of T1D patients, patients with onset of T1D, and patients at risk of developing T1D. Their presence overlapped with or preceded AAbs, and the extent of HERV-W-*env* expression appears to be correlated with disease progression. In β-cells, HERV-W-Env inhibits insulin secretion, possibly through its interaction with TLR4, which could also lead to decreased functionality and viability of β-cells. This is supported by the fact that downstream TLR4 signaling elements, such as NF-κB, MyD88, and TRIF, are upregulated in T1D patients [59]. Understanding their involvement in complex pathological disorders made HERV Env proteins potential targets for therapeutic intervention [45]. Of note, a monoclonal antibody directed against a HERV-W Env showed neuroprotective effects in progressive MS patients [45,60,61]. Another candidate susceptibility gene is the HERV-K18 locus on chromosome 1, which has characteristics of T-cell SAg. At the onset of T1D, patients have T cells with the reactive HERV-K18 SAg receptor in the endocrine pancreas, spleen, and blood. HERV-K18 might predispose to autoimmunity in a similar manner to HERV-W [59,62,63].

## 6. Technologies for the Study of Retrotransposons

Retrotransposons are an integral part of cell transcriptional output. They represent important sites of chromatin regulation and contribute to cell heterogeneity. A better understanding of the activity of retrotransposons in individual cells using single-cell transcriptomics will lead to a deeper understanding of cellular functions [64]. Specific retrotransposon types are expressed in subpopulations of embryonic stem cells and dynamically regulated during pluripotency reprogramming, differentiation, and embryogenesis. Retrotransposons are also expressed in somatic cells, including human disease-specific retrotransposons that are undetectable in bulk genomic and transcriptomic analyses [65]. In addition, the combination of long-read and short-read sequencing technology in single-cell RNA sequencing (scRNA-seq) is a powerful analysis technique to study retrotransposon sequences in cell-specific transcriptomes [64,66]. Single-cell sequencing is widely used to reveal the diversity and specificity of the immune cell repertoire in various diseases, from cancer to viral infections and autoimmunity. In addition, it is also useful for determining differential gene expression among immune cells to gain deeper insights into the intracellular regulation of the immune system [67]. 

The scRNA-seq has been used to characterize lymphocyte phenotypes in T1D in peripheral blood, pancreatic lymph nodes, and islets, and has revealed specific genes that are differentially expressed in islet-specific T cells in T1D. The scRNA-seq has also uncovered broader gene expression patterns involved in T1D, novel cell subtypes, and can predict T1D development before AAb are produced [67].

## 7. Best Practices for Analysis of Genomic Regions including Retrotransposons

Detection of retroviral integrations (not included in the human reference genome sequence) from genomic data is bioinformatically challenging [68]. As copy numbers of LINEs, Alus, and SVAs are actively increasing at an estimated rate of about two to five new insertions per 100 births for Alu, and about 0.5–1 in 100 for LINE -1, it stands to reason that most of these element insertions are not present in the reference genome assembly and are detectable as structural variants [41]. In addition, repeats often insert into other repeats, and often only fragments within fragments are present. Complete elements are very rarely found. Moreover, genome coverage is generally between 90% and 95%, with repetitive regions accounting for the unsequenced portion [69]. To overcome these challenges, de novo assembled personalized genomes would be best but have been proven difficult to implement. On the other hand, trying to identify highly repetitive regions and search for retroviral integration sites remains a viable option.

Repeat identification and masking is usually the first step in the computational phase of genome annotation and should be part of any genome annotation project [69]. To cope with the genetic complexity, as well as the unique EVE insertions repertoire of each genome, users usually need to assemble repeat sequences for their own genome of interest, and a personalized reference genome can be constructed [69,70,71]. This is quite a challenge because the sequences of retrotransposons are constantly changing and highly fragmented [70]. The tools available for creating a repeat library can generally be divided into two classes: homology-based and de novo tools. The masking step signals downstream sequence alignment and gene prediction tools for these repeat regions [69]. 

Homology-based and structure-based detection of inserted retrotransposons relies on several features of specific classes of retrotransposons, as well as features associated with their insertions. ERV elements can be identified by searching for LTRs (~100–1000 bp direct repeats) and characteristic motifs/protein domains (*gag*, *env*, and *pol*, including pathogenesis-related domain, reverse transcriptase, and endonuclease) [70]. The consequence of a transcription process and subsequent insertion is splice junction between coding exons [41]. On the other hand, the sequence structure observed in non-LTR retrotransposons is polyadenylation at the 3’ end [41]. The next insertion feature is related to reverse transcription of LINEs with ORF2 endonuclease cleavage, which results in target site duplication (TSD), i.e., the sequence appears between the upper and lower strand cuts (typically 7–20 bp) on both sides of the new insertion. Finally, perhaps the most important feature of retrotransposon insertions that affect the methods used to detect them is their repetitive nature [41]. RepeatMasker [50] identifies sequence segments in a target genome that are homologous to known repeats. However, there are thousands of elements from each active class of EVEs in the human genome, and this is the key factor that complicates the accurate detection of EVE insertions because read will possibly map to various locations in the reference genome [41]. In addition, large insertions or deletions that occur in EVEs after they have been integrated into the genome may cause detection methods to identify two matches to a query sequence rather than one match with a long gap. Therefore, a post-processing step called “defragmentation” is often required to assemble fragments of a single EVE insertion event into a biologically meaningful annotation. For example, the ProcessRepeats script distributed with RepeatMasker links LTRs to the body of the corresponding LTR retrotransposon and links poly(A) sequences to the tail of non-LTR retrotransposons [72].

De novo repeat identification has the advantage that it can be used to identify EVE families that do not belong to a known class or do not share one or more of the characteristic features [70]. The product of de novo programs is therefore an as complete as possible library of reconstructed TEs. However, de novo tools are not specific to TEs, so their outputs may also include tandem repeats, segmental duplications, and satellites, as well as protein-coding genes, such as histones and tubulins [69,72]. There are several public databases of EVEs and a number of programmes for analysing repeat sequences. The TEHub project is an open and collaborative platform for researchers interested in the diversity, identification, and annotation of transposable elements, and also provides a list of tools for the analysis of EVE [70,73]. For example, dbRIP is a database of human Retrotransposon Insertion Polymorphisms (RIPs) in which RIPs are highly integrated into the genome annotation data provided by the UCSC Genome Browser. dbRIP contains all currently known polymorphic Alu, LINE-1, and SVA insertion loci [74,75]. Repbase is another repetitive sequence database used as a reference for annotating the presence of repetitive DNA [76,77]. Pačes et al. [78] collected copies of known ERVs in Repbase Update [79] and designed the HERVd database. It can be used to search for individual HERV families, identify HERV parts, graphically output HERV structures, compare HERVs, and identify retrovirus integration sites [80,81]. In addition, HERVd provides links to other databases [82]. For HERV enrichment analysis, there is a bioinformatics tool EnHERV that provides the functional characteristics of HERVs [46,83]. However, there is still no consensus on the EVEs subfamilies that remain active in the human genome. Therefore, Autio et al. created a catalogue of recently mobile subfamilies (RMS) that excludes many false-positive results in public databases. The catalogue RMS is a valuable resource in the search for a possible disease-causing factor in disorders of unknown aetiology.

## 8. Conclusions and Perspectives

With the advancement of sequencing technologies and computational tools, the study of the retrotransposon association with physiology and diseases is becoming a hot topic among researchers. In particular, multifactorial diseases, such as T1D, are of interest in examining the involvement of retrotransposons in disease aetiology. The identification of novel factors that trigger disease onset may represent promising new therapeutic targets for patient-centered interventions to improve quality of life. Combinations of immunotherapeutic agents targeting different types of immune responses may have greater potential to induce durable remission. Analyses of some interventions have shown that response to therapy varies widely among patients. Well-established immune and metabolic biomarkers for T1D (autoantibodies and C-peptide) may be too robust. Therefore, further knowledge of the underlying pathophysiological mechanisms may be important to adequately define subgroups of patients. Some HERV proteins have already been associated with various neurodegenerative and autoimmune diseases as well as cancer. However, the coding retrotransposon sequences represent only a minority of millions of EVE-derived fragments that could contribute to genetic instability and thus disease progression. When and why these mobile elements are activated remains a central question. Environmental stimuli, including certain infections, have been associated with the activation of retrotransposons. Inclusion of the sequences of EVEs, which make up nearly half of the human genome, in the reference genome would be of great importance to scientists studying EVEs.

## Figures and Tables

**Figure 1 ijms-24-03104-f001:**
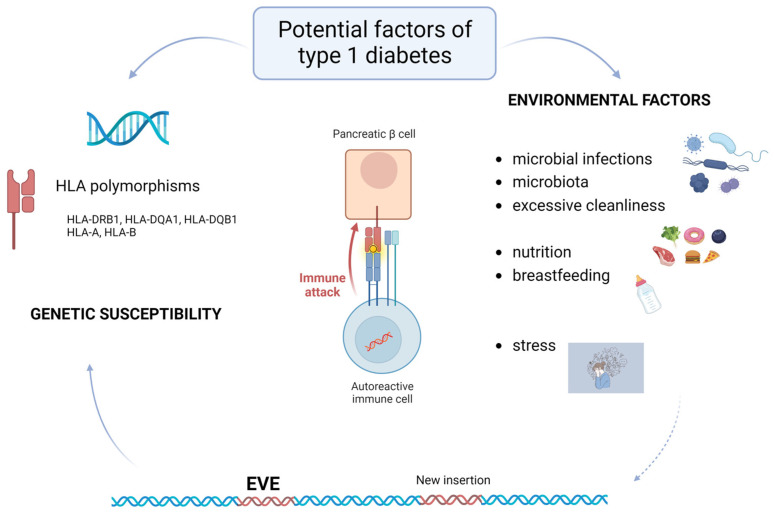
Interplay of factors associated with type 1 diabetes (T1D). Solid arrows indicate established connections, while the dashed arrow indicates a hypothesized connection. HLA—human leukocyte antigen, EVE—endogenous viral element.

**Figure 2 ijms-24-03104-f002:**
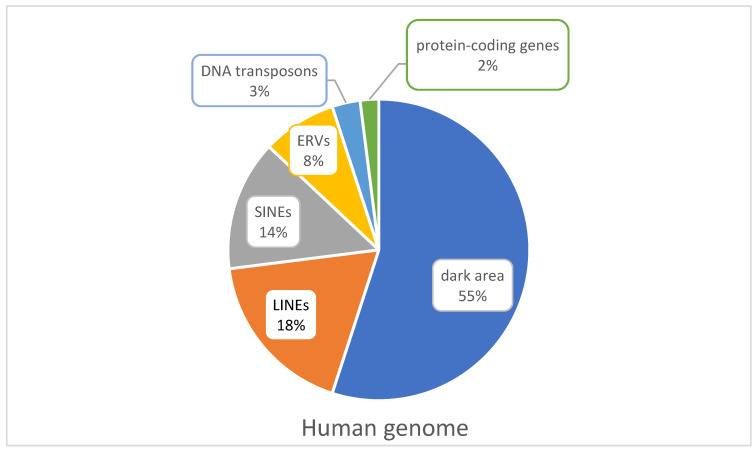
Relative proportions of different types of DNA sequences in the human genome. LINEs—long interspersed nuclear elements, SINEs—short interspersed nuclear elements, ERVs—endogenous retroviruses.

**Figure 3 ijms-24-03104-f003:**
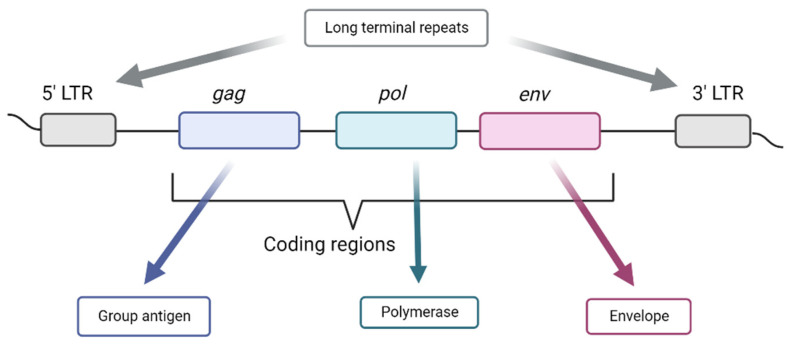
Main components of a human endogenous retrovirus (HERV). LTR—long terminal repeat.

**Figure 4 ijms-24-03104-f004:**
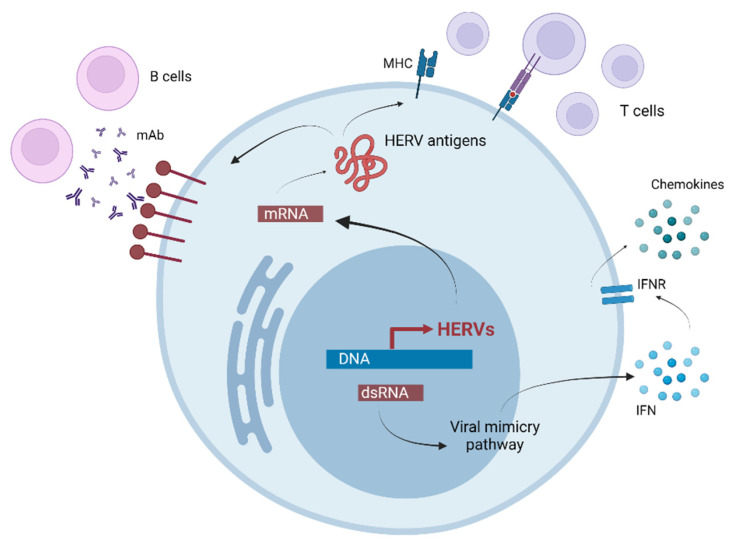
Some pathways of active HERVs. HERV—human endogenous retrovirus, MHC—major histocompatibility complex, mAb—monoclonal antibody, IFN—interferon, IFNR—IFN receptor.

**Table 1 ijms-24-03104-t001:** Important human leukocyte antigen (*HLA*) loci regarding T1D and retrotransposons of selected families in 6p21 genome region.

HLA Locus	Alu Family	LINE-1 Family	ERV Family
DQA1	AluJr, AluJo		MER51A, MER51B
DQB1	AluYh3		LTR13
DRB1	AluYa5, AluJb, AluSc8, AluSq2, AluSg, AluSx	L1PA3, L1PA4, L1P1, L1M5	LTR12
B	AluJr, AluJb, AluSq2, FLAM_A, FLAM_C	L1M4a1, L1M4a2, L1M5, L1ME1, L1MEf, L1MD, L1PA8A, L1PA13, HAL1	

## Data Availability

Not applicable.

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
