# Peer review of "Contribution of Retrotransposons to the Pathogenesis of Type 1 Diabetes and Challenges in Analysis Methods"

_ijms, 2023, doi:10.3390/ijms24043104_

Round 1

Reviewer 1 Report

I read with interest a manuscript entitled "Contribution of Endogenous Viral Elements to the Pathogenesis of Type 1 Diabetes and Challenges in Analysis Methods" written by Štangar et al. in which they discussed the possible involvement of viruses and endogenous viral elements (EVEs)  in type 1 diabetes pathogenesis. 

The introduction is well organized and the authors started with information about type 1 diabetes and how different environmental, including viruses, and genetic factors could have a major role in that. Then they discussed the EVE involvement in different pathological conditions, including T1D. 

In the second section entitled "Reported Relationship between the Occurrence of T1D and Viral Infections", the authors reported some studied mechanisms and examples of viruses that could contribute to T1D. Plus, the authors mentioned EVs which are widely studied in T1D pathogenesis, then the authors described Rotavirus and SARS-CoV-2 and ended with hygiene theory which could explain some aspects of T1D pathology. In the third paragraph, the authors defined EVEs and their types, characteristics and how they could be involved in diseases. Then the authors discussed the influence of EVEs on diseases and on T1D specifically. Then the authors presented the technical aspects of EVEs and the challenges associated with the available techniques. 

To improve this review, the authors are recommended to consider the following:

For the first section:

# Line 37-39: more recent data are available and seem quite different from what is written here.

# Line 39-42: This sentence must be rewritten.

# Line 63: Beta in greek sounds more scientific than in letter, especially as it is presented in greek in Figure 1.

# A sentence about the clinical feature of this disease and how it's different from T2DM is recommended to attract readers with a clinical background. 

For the second section

# Line 83: Beta to be changed to the greek letter

# Line 113: NO is mentioned as an abbreviated term, it must be written in full for the first time. 

In the fourth section

# My only suggestion in this part is that the authors can make the section on the influence of EVEs on T1D a new section

Conclusion and perspectives 

# I would like to read how this could help us to tackle T1D and any therapeutic and preventive measures that could be used based on the current understanding or as a future direction. 

Reviewer 2 Report

I have reviewed the paper by Stangar et al.

The topic is interesting, but needs a better and more logical flow.

What is the rationale behind moving from HLA to EVEs in the Introduction. A better segway and a clear rationale to connect EVE with T1D is needed before presenting things just as “…In T1D they may represent the missing link between genetic susceptibility 71 and environmental factors (Figure 1) [18]. Therefore, in this review,…”

The section on viral infections has really no relationship with EVE. An immune response is mounted again an infecting viral organisms. Then they introduce SARS-CoV-2 which does not integrate nor produces cDNA, and ends up talking about the immune response in COVID-19.

EVE are integrated genes (as the authors point our “EVEs can cause disease by two primary mechanisms: insertional mutagenesis and 131 chromosomal rearrangements”). Again, the connection here is not well made.

It sounds much like a transposon mechanism, yet the authors decide not to deal with these elements. They deep only with retrotransposons.

Then call the paper The role of retrotransposons…rather then EVEs.

Last part of Section4, does not tell any mechanistic model. Just a series of genetic epidemiological findings, listed after the same vague verbiage “HERVs are an important group of viral entities, which may be associated with T1D 257 and other autoimmune diseases etiology. HERVs have been shown to induce the produc-258 tion of pro-inflammatory cytokines such as IL-1, IL-6, and TNF-α by cells such as T1D 259 triggering monocytes [18,52]. They represent a potential link between genetic and environ-260 mental factors and can be transactivated by environmental viruses such as enteroviruses, 261 by some bacteria, and by inflammatory stimuli [2,18].”

Section 5 and 6 are totally unnecessary. Plus they are excessively long.

Round 2

Reviewer 2 Report

All of my concerns and suggestions have been adequately addressed by the authors.

Manuscript is acceptable for publication.